# Pressurization Ventilation Technique for Controlling Gas Leakage and Dispersion at Backfilled Working Faces in Large-Opening Underground Mines: CFD Analysis and Experimental Tests

**Van-Duc Nguyen** [1,*], **Won-Ho Heo** [2], **Rocky Kubuya** [1] **and Chang-Woo Lee** [1,*]

[1]  Department of Energy and Mineral Resources, College of Engineering, Dong-A University, Saha-gu, Busan 49315, Korea; pakrockykiro@gmail.com
[2]  Mining Tech Co., Ltd, Saha-gu, Busan 49315, Korea; hoya4016@naver.com
*  Correspondence: nguyenduc.imsat@gmail.com (V.-D.N.); cwlee@dau.ac.kr (C.-W.L.)

**Abstract:** Pressurization ventilation techniques, originally designed to control building fires, have never been applied to the mines. The working face, backfilled with fly-ash-based materials, is likely to be contaminated during, and even after, the curing period of the backfill materials. Gases such as $NH_3$ and $CO_2$ may leak out prolongedly from the backfilled sites. Proper ventilation schemes should be implemented to control toxic gas leakage and thus minimize the workers' exposure. This study aims at evaluating the applicability of a pressurization ventilation scheme at backfilled working faces in large-opening limestone mines. To pressurize the working face, two different fans (15 kW and 37 kW) were developed and two ventilation scenarios were tested. Computational Fluid Dynamics (CFD) analysis was also carried out for comparison purposes. There is no established standard for differential pressure between the inside and outside of working faces to prevent gas leakage at mines. However, taking the differential pressure of 50 Pa in British standards for controlling building fires (where a relatively stronger dissipation force than the gas leakage of a mining face occurs), the pressure differential created by two blowing fans seems to be sufficient to control the gas leakage and dispersion within the work space.

**Keywords:** pressurization system; ventilation fan; differential pressure; gas leakage; backfilled area

## 1. Introduction

Backfilling of voids generated by mining is a fundamental component of most underground stopping operations. Without backfill, these operations would be unsafe, less productive, and shorter in life [1]. It is expected that large quantities of $NH_3$ and $CO_2$ will leak into the atmosphere, as well as the working space, in the mining area backfilled with the composite carbonate-based material. At most thermal power plants, a large amount of NH4 is injected into the denitrification unit of the flue gas cleaning system, thereby reducing NOx to $N_2$ and water through catalytic reduction. As a result, NH4 is retained in the fly ash bulk and is likely to leak out for a long time. This is why some countries, such as Poland, regulate NH4 levels in the air at backfilled mine sites. Therefore, it is necessary to control the thermal environment and the air quality within the underground working space during, and even after, the curing period for backfilled materials. In large-opening underground limestone mines in Korea with typical airway dimensions of 10 m (W) and 8 m (H), the conventional ventilation system of large-capacity axial-flow fans with ducts may be insufficient to control the toxic gas which escapes during the curing period, since those mines are developed with multiple interconnected levels and the intake and return airways are not always well-maintained.

Thus, the ventilation system proposed in this study has the twofold purpose of minimizing the ventilation cost while isolating the contaminated zone near the vicinity of the face in order to reduce the workers' exposure. To achieve this goal, we suggest the implementation of the pressurization ventilation system which was originally designed to control building fire dispersion and secure safe escape routes by generating a pressure difference of at least 50 Pa between the escapees and the fire-affected space. The same concept is applied to the backfilled site to generate a pressurized zone just in front of the face to minimize the leakage and confine the polluted air within the pressurized zone. Therefore, the ultimate goal of this study is to secure a safe working environment at backfilled sites by applying a low-cost pressurization ventilation technique.

## 2. Pressurization Ventilation Technique

### 2.1. Background of the Pressurization Ventilation Technique

Pressurization ventilation techniques were developed for the purpose of creating safe zones within burning buildings for a variety of purposes, like maintaining escape, and firefighting access routes in lobbies and stairwells, and creating refuge areas for trapped victims. The aim of the pressure differential system is to establish airflow paths from protected spaces at high pressure to spaces at lower or ambient pressure, preventing the spread of toxic gas released during a fire [2]. The successful operation of the pressurization system remains a standard feature of high rise building codes in USA, UK, Australia, China, India, the UAE, and many other locations. A pressurization system applied in buildings is intended to prevent smoke leaking past closed doors into stairwells by injecting clean air into the stair enclosure at sufficient pressure that the air pressure in the stairwell is greater than that in the adjacent fire compartment [3]. Studies of pressurization systems by Tamura 1989, 1992 [4,5], Budnick 1987 [6], Wang et al. 2004 [7], Jo et al. 2007 [8], Gai et al. 2017 [2], and Li et al. 2018 [9] form the theoretical foundation for the development of ventilation schemes for gas leakage and dispersion controlling at backfilled underground mine sites. Figure 1a illustrates the basic concept of a pressure differential system for generating and maintaining safe zones within buildings. Generally, the positively pressurized zone can be generated by the operation of a fan in an enclosed space. The purpose of a positively pressurized compartment is to prevent the spread of smoke from the compartment of first ignition to other areas. Thus, during a fire, a refugee can move to a high-pressure compartment for safety.

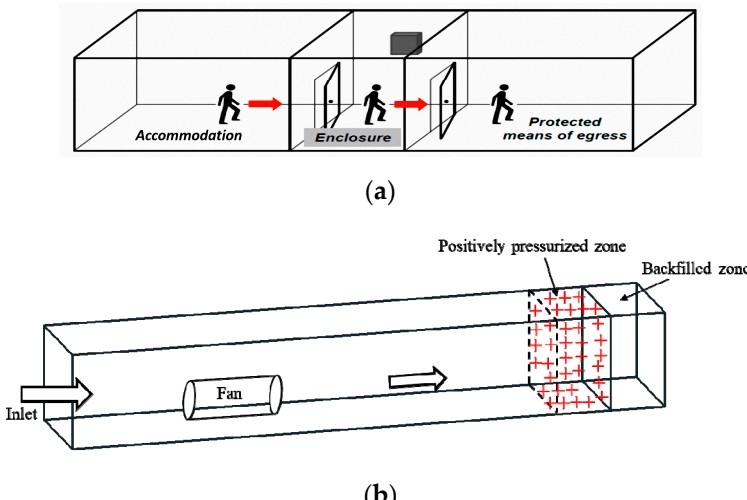

(**a**)

(**b**)

**Figure 1.** Background of pressurization ventilation technique. (**a**) Basic concept of the pressure differential system [2]; (**b**) application of the pressurization ventilation concept at a backfilled area.

Figure 1b shows the application of the pressurization ventilation technique at a mine working space. It can be seen that a positively pressurized zone near the backfilled area is be generated by blowing fans. The pressurized zone is expected to prevent hazardous gas leakage from the backfilled zone. According to the British standards BS EN 12101-6 "Specification for pressure differential systems" [10] and BS 5588-4 "Code of practice for smoke control using pressure differentials" [11], there are two requirements to maintain a pressurization system. Firstly, the minimum pressure differential between inside and outside of the pressurized zone is 50 Pa. Secondly, for an open door condition, the airflow discharged from the pressurized zone through the doorway should be not less than 2.0 m/s.

## 2.2. Description of Fans for the Pressurization Ventilation System

Several studies at the National Institute for Occupational Safety and Health (NIOSH) by Grau et al. 2002 [12,13], Krog et al. 2006 [14,15], and Chekan et al. 2006 [16] indicated that large diameter, low-pressure fans provide better regional air coverage in large-opening mines than high-pressure fans. However, while low-pressure fans installed in a large-opening airway can generate a relatively large amount of slow-moving airflow at a low cost, the pressure differentials created by those fans are not sufficiently high. This limitation is due to the fan locations; fans must be installed at a distance far enough away from the working face to allow space for the many equipment employed there. Thus, for the pressurization purpose at backfilled areas in large-opening mines in this study, two types of fans were developed—a 37 kW high-pressure fan and a 15 kW low-pressure fan. Figure 2 shows the two axial-flow high-pressure and low-pressure fans used for the pressurization ventilation experiment in this study. Their specifications are summarized in Table 1.

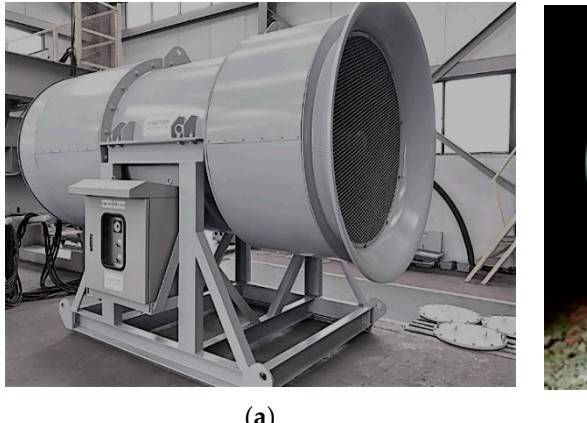 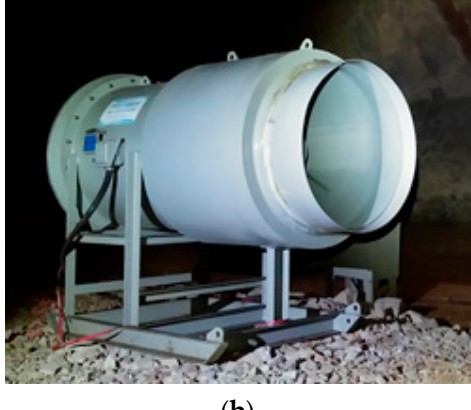

(**a**)          (**b**)

**Figure 2.** Two fans developed for the pressurization ventilation experiments: (**a**) 37 kW high-pressure fan; (**b**) 15 kW low-pressure fan.

**Table 1.** Fan specifications.

| Categories | 37 kW High-Pressure Fan | 15 kW Low-Pressure Fan |
|---|---|---|
| Flow quantity (m$^3$/s) | 47.1 | 16.67 |
| Diameter (m) | 1.4 | 0.95 |
| Pressure (Pa) | 555.1 | 235 |
| Discharge velocity(m/s) | 30.6 | 23.5 |
| Noise level (dB(A)) | 105 | 103 |
| Power (kW) | 37 | 15 |
| Length (m) | 3.0 | 2.23 |
| Weight (kg) | 998 | 792 |

## 3. Mine Site Study of the Pressurization Ventilation Technique

### 3.1. Description of Mine Site Study and Experiment Method

In Korea, most underground limestone mines have been developed through the multi-level room-and-pillar mining method with large openings in steeply-dipping and faulted ore bodies. In general, the entries are 6–9 m high and 10–15 m wide and the rampways connecting levels with a vertical difference of approximately 20 m are inclined at 10%–13%. The number of entries in each level depends on the vein width; two to five entries being typical. Figure 3a shows a 3D map of a large-opening underground mine, the D limestone mine in Chungbuk province. Pressurization ventilation experiments were carried out at a blind entry development site in Level 5; the development entry was 8 m (W) × 7 m (H) and the experiment segment was 188 m long as shown in Figure 3b.

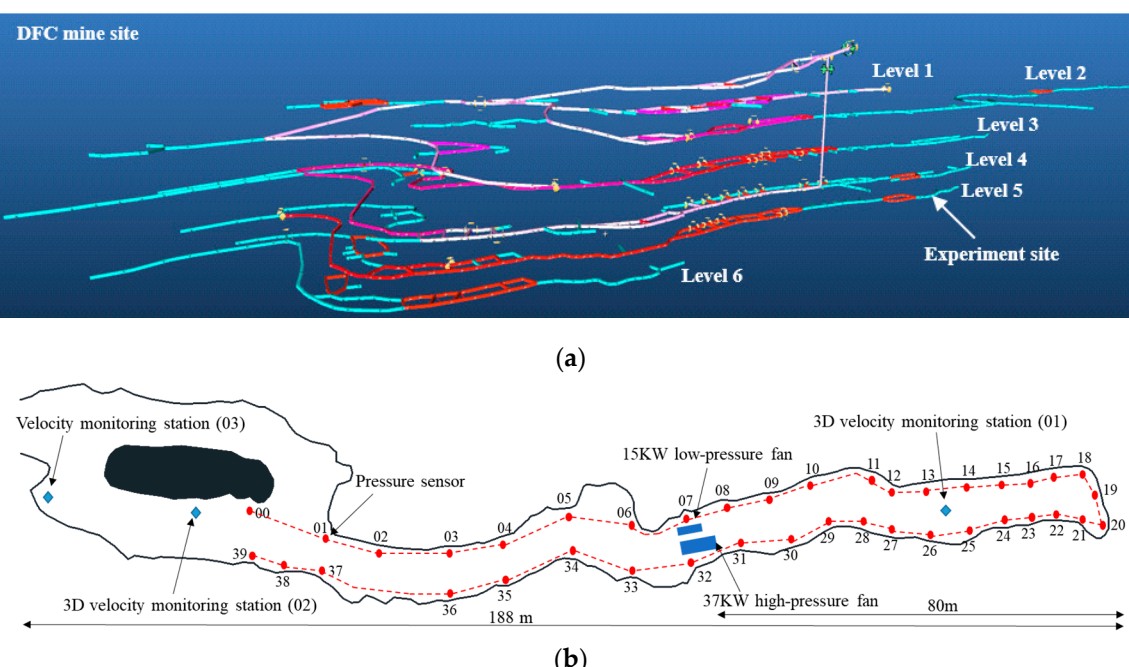

(a)

(b)

**Figure 3.** D large-opening mine and layout of pressurization ventilation experiment. (**a**) D limestone mine in Chungbuk province; (**b**) layout of the experiment site at the fifth level.

Figure 3b shows the locations of the test equipment installation and monitoring stations. Two pressurizing ventilation fans were installed 80 m from the working face, a hypothetical backfilled working face. The 37 kW high-pressure fan was installed on the right side of the entry facing toward the workface, while the 15 kW low-pressure fan was installed on the opposite side. Both fans were installed in the blowing mode. At the same time, the 40 pressure sensor modules, as shown in Figure 4, developed for this study were installed along the side of the entry, as shown in Figure 3b, to monitor pressure changes during the tests. A controller area network (CAN) was employed as the pressure sensors' communication method. CAN is the international standardization organization-defined serial communications bus originally developed for the automotive industry to replace complex wiring harnesses with a two-wire bus [17]. Pressure distributions during the fan operation were monitored and the results were analyzed to identify the existence of the pressurized zone near the face. In addition, three velocity monitoring stations were installed to evaluate the exhaust efficiency of contaminated air to the areas outside of the working face, as shown in Figure 3b and Table 2.

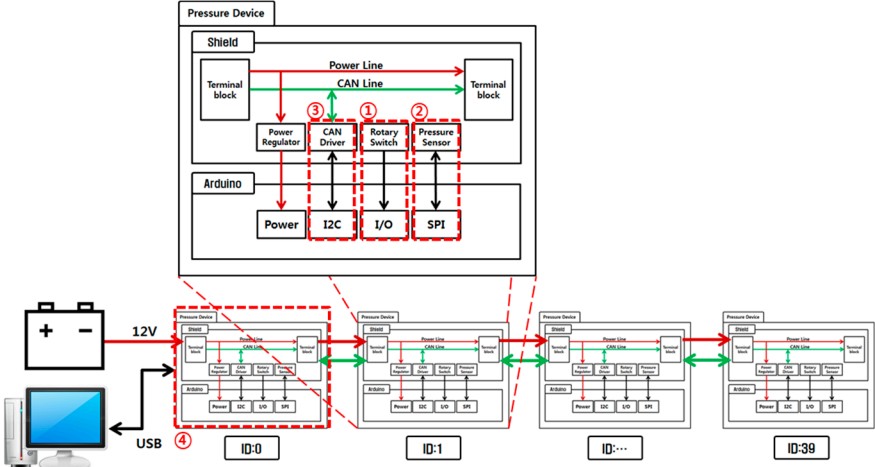

**Figure 4.** Pressure monitoring and communication module. CAN: controller area network.

**Table 2.** Experimental and numerical scenario description.

| Experiment Scenarios | Airway Layout | Description |
|---|---|---|
| Scenario I | 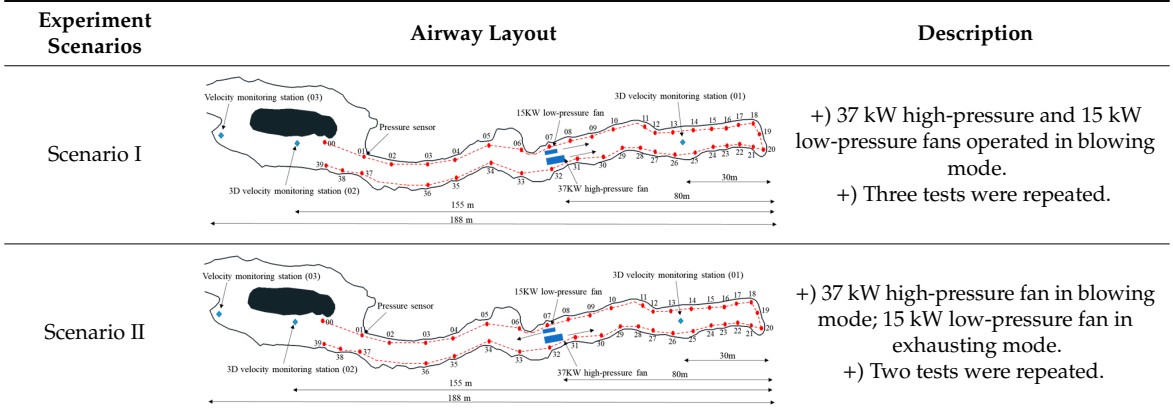 | +) 37 kW high-pressure and 15 kW low-pressure fans operated in blowing mode. <br> +) Three tests were repeated. |
| Scenario II | 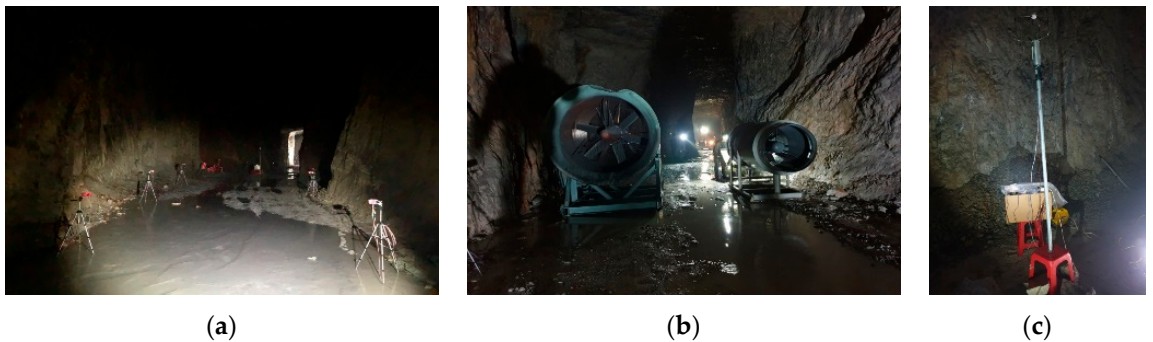 | +) 37 kW high-pressure fan in blowing mode; 15 kW low-pressure fan in exhausting mode. <br> +) Two tests were repeated. |

Figure 5 shows the pressure sensors (a) and fans (b) installed at the test site. In addition, three velocity monitoring stations were set up (c). Two experimental scenarios of the pressurization ventilation technique application described in Table 2 were evaluated. In Scenario I, both fans were operated in blowing mode, while in Scenario II the 37 kW fan was operated in blowing mode and the 15 kW fan was operated in exhaust mode.

| (a) | (b) | (c) |
|---|---|---|

**Figure 5.** Fans, pressure sensors, and 3D velocity monitoring installation in experiment site. (**a**) Pressure sensors; (**b**) fans; (**c**) velocity station.

*3.2. Experimental Results*

3.2.1. Scenario I: 15 kW and 37 kW Fans in Blowing Mode

　　Figure 6 shows the pressure data collected by 40 pressure sensors during the Scenario I tests. The first test was conducted for 49 min, while the second and third tests were carried out for 60 min. As shown in Figure 6a,c,e, the pressure data show that the initial pressure right at the beginning of fan operation increases dramatically and then the pressures at all sensor locations fluctuate. However, the pressure differentials among all the sensor locations remain almost constant. As clearly observed in Figure 6, the pressure data were divided into two zones—the outlet side of the fans showing higher pressures and the inlet side of the fans showing lower pressures. At all sensors in the outlet side (nos. 8–31), relatively higher pressures were monitored, while the sensors showing lower pressures (nos. 0–6 and 32–39) were located on the inlet side. The same trends were observed in all three tests as described in Figure 6a,c,e; the pressure differentials between those two groups of pressure data sensors remained constant even with serious fluctuations of the pressure within the working space due to the fan operations. This result indicates that a positively pressurized zone can be created and maintained continuously near the face during the fan operation to prevent gas leakage from the backfilled zone.

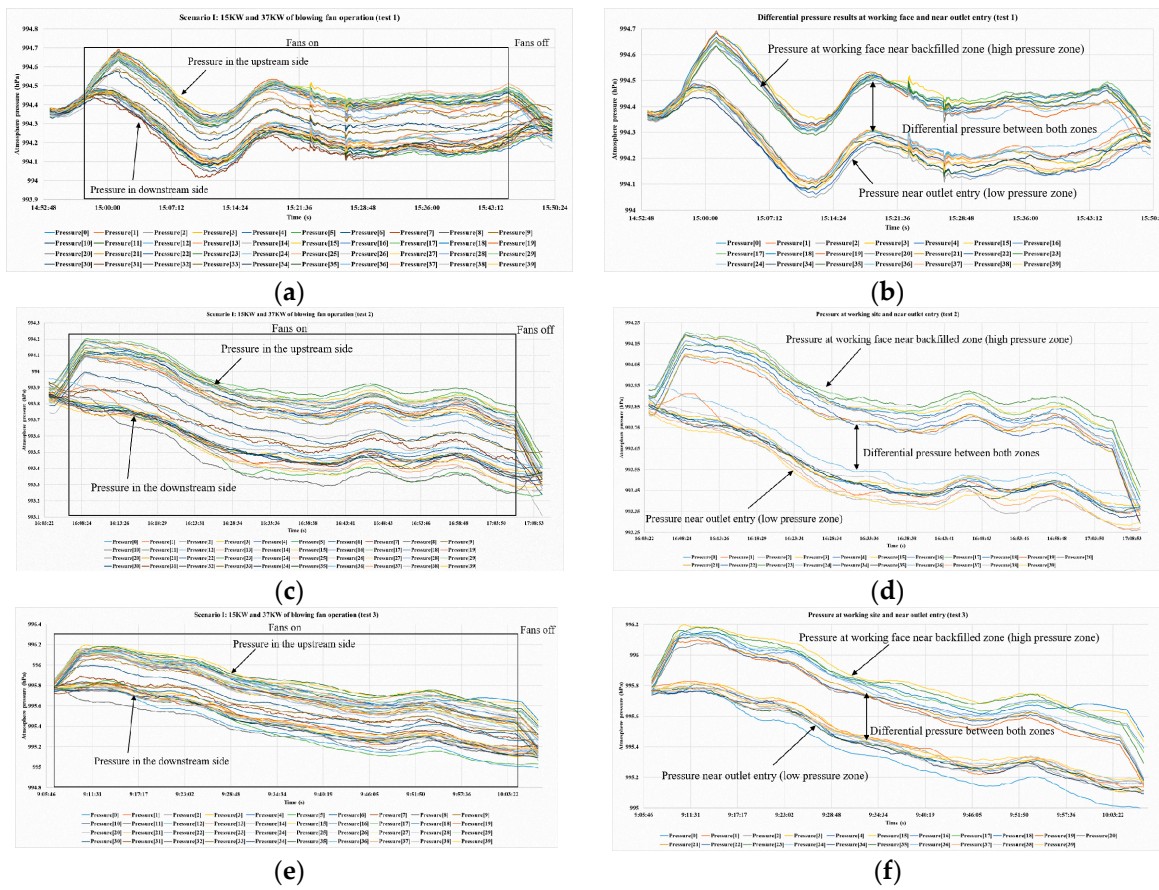

**Figure 6.** Temporal pressure distributions for Scenario I. (**a**) Test 1—49 min of fan operation; (**b**) differential pressure in high- and low-pressure zone (test 1); (**c**) test 2—60 min of fan operation; (**d**) differential pressure in high- and low-pressure zone (test 2); (**e**) test 3—60 min of fan operation; (**f**) differential pressure in high- and low-pressure zone (test 3).

　　Figure 7a,c,e show the average pressures of these two groups and the pressure differentials between them. The differential pressure between the outlet and inlet sides of the fan was 22.30 Pa in the first test, while the second and third tests showed differences of 32.78 Pa and 30.50 Pa, respectively. However, considering the differential pressure of 50 Pa specified in the British standards of BS EN

12101-6 and BS 558-4 to control building fires, which has relatively larger dissipation force than the gas leakage in the mining face, the pressure difference of approximately 30 Pa created by two blowing fans in this study may be sufficient to control the gas leakage and dispersion within the working space. Further research is planned for determining the minimum level of pressure differential required to control the gas leakage at the backfilled sites.

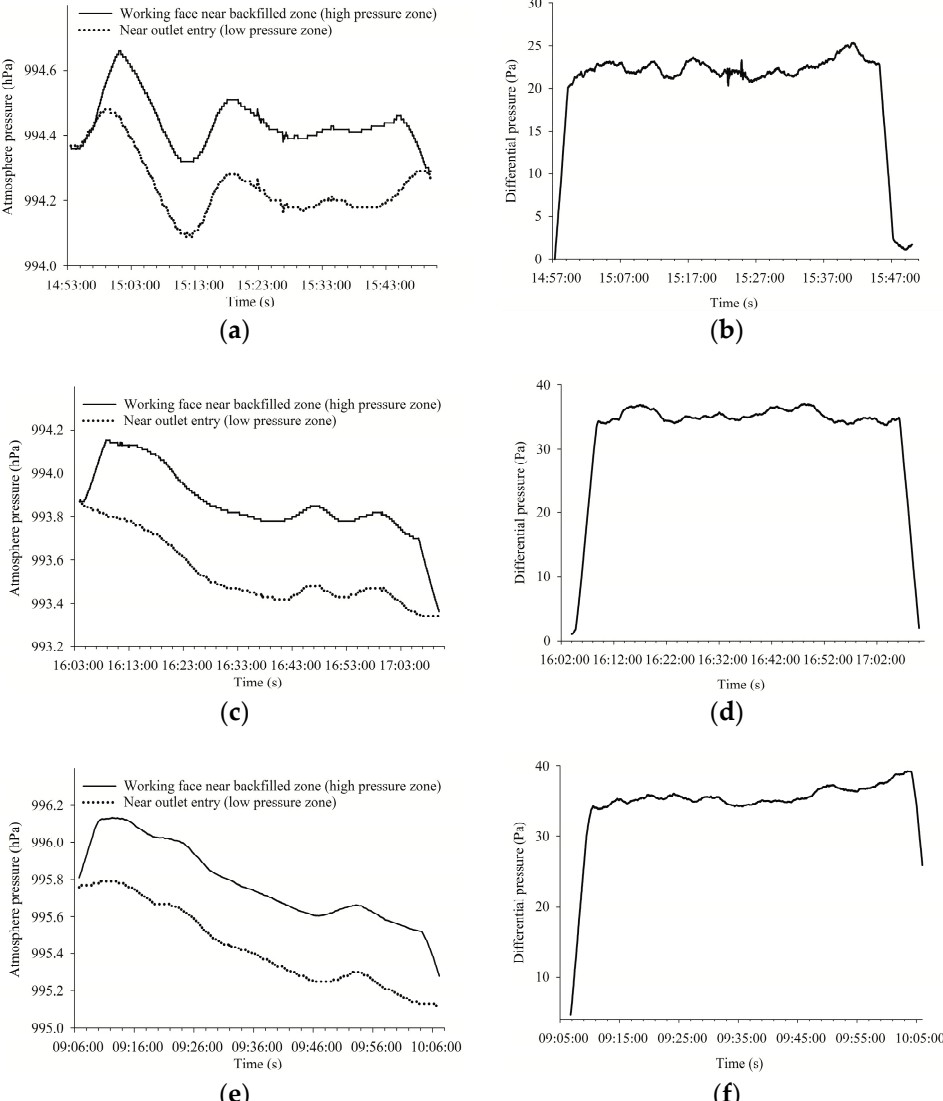

**Figure 7.** Differential pressure at high- and low-pressure zone in Scenario I. (**a**) Average differential pressure (test 1); (**b**) differential pressure in high- and low-pressure zone (test 1); (**c**) average differential pressure (test 2); (**d**) differential pressure in high- and low-pressure zone (test 2); (**e**) average differential pressure (test 3); (**f**) differential pressure in high- and low-pressure zone (test 3).

The monitored pressure data were rearranged in Figure 8 to show the spatial distribution from the face to the end of the experiment space. During three tests of scenario I with two blowing fans operational, positive high-pressure zones can be observed within a distance of 40 m from the working face, while in the inlet side between 60 and 140 m, the low-pressure zone can be seen clearly. This result indicates that with the fan operation in blowing mode, a positive high-pressure zone can be generated continuously near the face.

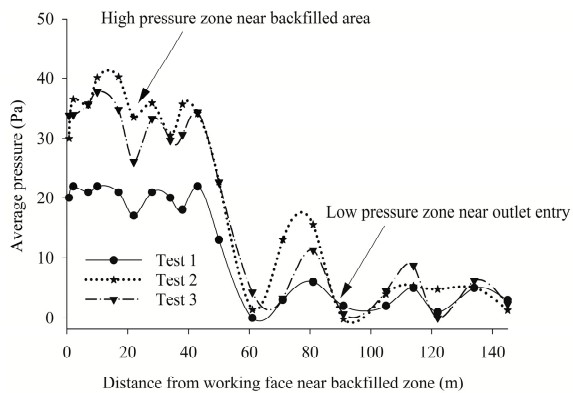

**Figure 8.** Differential pressure by distance in Scenario I.

Figure 9 shows the velocity distribution measured at the velocity monitoring Station 2. During the fan operation, the velocity of the airflow moving out of the working space was kept under 0.01 m/s; this implies that the air was circulating only within the pressurized working space and that contaminants generated were also expected to remain within the pressurized zone.

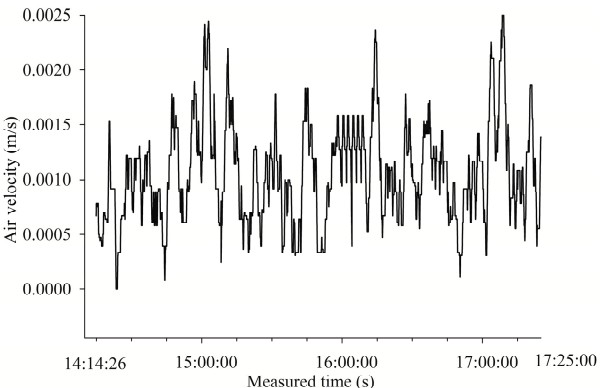

**Figure 9.** Velocity profiles at velocity monitoring Station 2.

### 3.2.2. Scenario II: 15 kW Fan in Exhausting Mode and 37 kW Fan in Blowing Mode

The test results for Scenario I with 15 kW and 37 kW fans in blowing mode show that the whole area down to the working face can be well pressurized to approximately 30 Pa higher than the opposite side of the fans and this pressurized zone can keep the toxic gases that have leaked from the backfilled face within the zone. In Scenario II, the 15 kW low-pressure fan was operated in exhausting mode and the 37 kW high-pressure fan in blowing mode. This scenario was designed to see if a certain portion of the contaminated air within the pressurized zone can be exhausted through the safe airflow path to reduce workers' exposure. Two tests were carried out for Scenario II. Figure 10 shows the pressure distributions monitored by 40 sensors. Figure 10a,c illustrate the recorded pressure data during 60 min of fan operation for the first test and 30 min for the second test. Compared to Scenario I, it can be observed that the high- and low-pressure zones are not well differentiated. However, as in Figure 6b,d,f, the pressure data from the sensors near the working face and near the left boundary of the experiment space were compared in Figure 10b,d. A similar pattern indicating the pressurization can be observed but the differential pressure between both zones is much smaller than in Scenario I. This result indicates that positive pressurization can still be created in this experiment, but the pressurization effect is not significant.

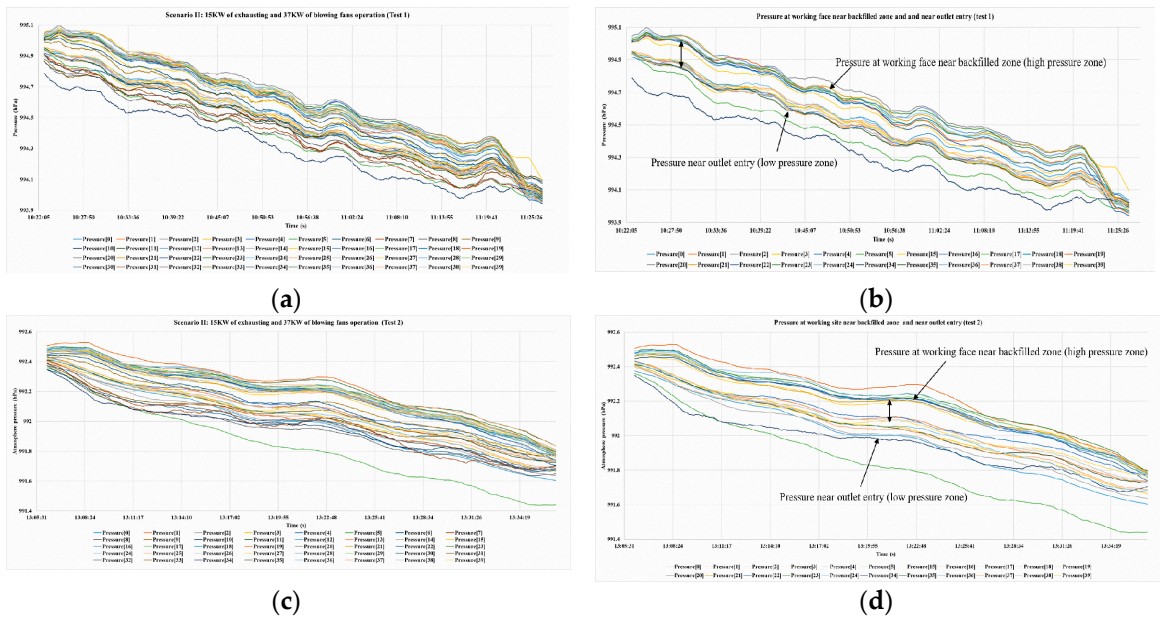

**Figure 10.** Temporal pressure distributions for Scenario II. (**a**) Test 1—60 min of fan operation; (**b**) differential pressure in high- and low-pressure zone (test 1); (**c**) test 2—30 min of fan operation; (**d**) differential pressure in low- and high-pressure zone (test 2).

The average pressures and the pressure differentials on both sides are plotted in Figure 11. In this regard, the high positive pressure zone located at the working face can be clearly observed in Figure 11c,d. The average pressure differentials of the two tests were 15.38 and 17.56 Pa, respectively. Thus, even with the 15 kW fan in exhausting mode, the working zone near the backfilled area can still be pressurized by only the 37 kW high-pressure fan in blowing mode. However, whether the pressure differentials created in Scenario II are enough to control the leakage and dissipation will be analyzed further in the following section.

Figure 12 shows the pressure distribution by distance. During two tests of scenario II, the high-pressure zone exists within 40 m from the working face, while within 60 to 140 m, the low-pressure zone can be located.

Based on Figure 13, which shows the velocity profiles at Station 2 during the fan operation, the airflow discharged by the 15 kW exhaust fan was almost stagnant with the average velocity less than 0.01 m/s. This implies that as in Scenario I, the contaminated air in the working zone can be hardly exhausted even with a fan in exhaust mode.

The test results for the two scenarios are summarized in Table 3. The results show that the pressure differentials on each side of the fans were well maintained, above 22 Pa in Scenario I and 15 Pa in Scenario II. This pressurized zone near the working face and the extremely low exhaust efficiency shown by two-fan operation implies the applicability of the pressurization ventilation technique to control gas leakage from the backfilled working face and dissipation of the polluted air.

**Table 3.** Summary of the experimental results.

| Categories | | Differential Pressure (Pa) | Average Velocity at Station 2 (m/s) |
|---|---|---|---|
| Scenario I | Test 1 | 22.30 | 0.001 |
| | Test 2 | 32.78 | 0.001 |
| | Test 3 | 30.50 | 0.001 |
| Scenario II | Test 1 | 15.38 | 0.012 |
| | Test 2 | 17.56 | 0.011 |

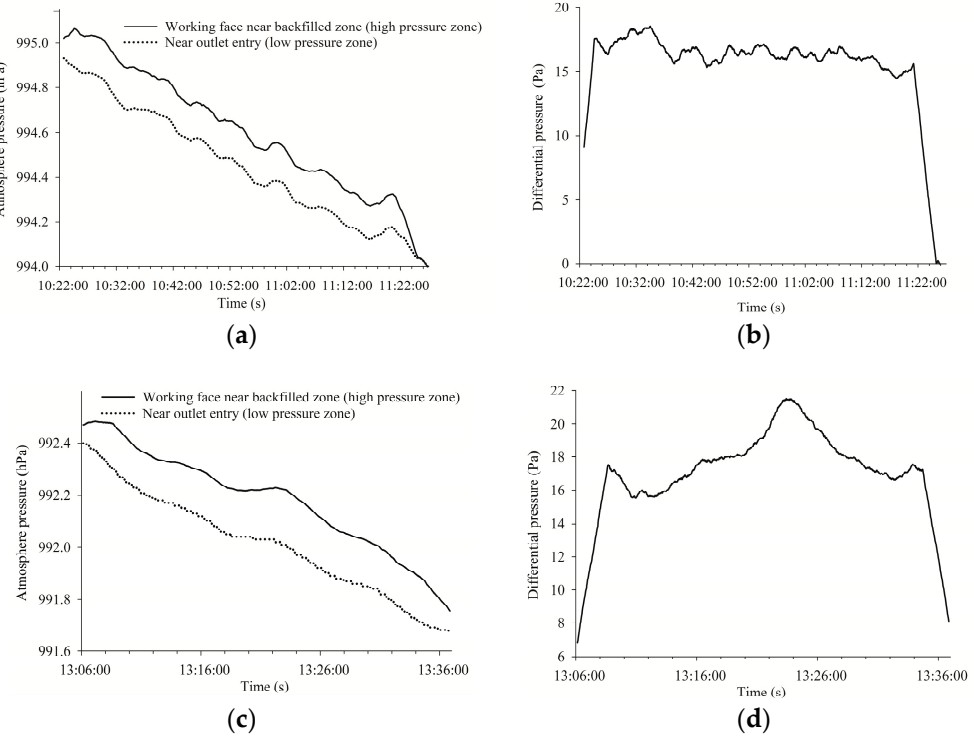

**Figure 11.** Differential pressure in high- and low-pressure zone in Scenario II. (**a**) Average differential pressure (test 1); (**b**) differential pressure in high- and low-pressure zone (test 1); (**c**) average differential pressure (test 2); (**d**) differential pressure in high- and low-pressure zone (test 2).

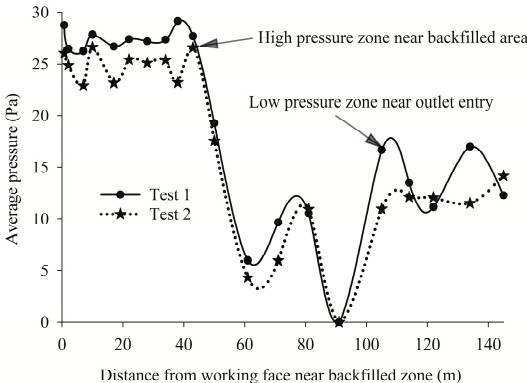

**Figure 12.** Differential pressure by distance in Scenario II.

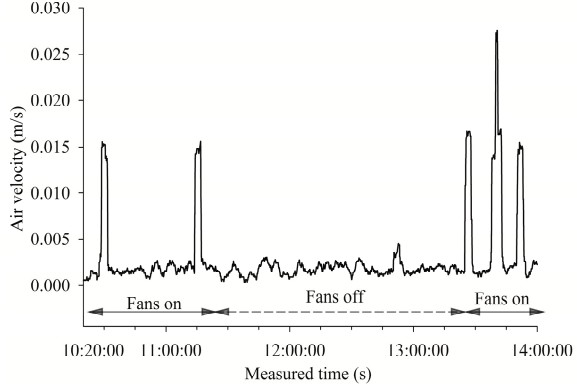

**Figure 13.** Velocity profiles at velocity monitoring Station 2.

## 4. Comparison through CFD Analysis

### 4.1. CFD Analysis

Applications of the CFD technique in the underground mine environment can be found in numerous studies. In this study, the CFD analysis was carried out using ANSYS FLUENT, which has commonly been used by researchers to study various fluid-flow and heat transfer problems in the underground mine environment [18]. Based on the real scale of the experiment site in Figure 3b, the 3D geometry layout for the CFD analysis is illustrated in Figure 14. The initial condition of the CFD simulation can be shown in Table 4. The $k - \varepsilon$ model was employed as the turbulence model in this study and all computational iterations were solved implicitly. The defaulted convergence absolute criteria were $10^{-4}$ for the continuity and momentum equation, and $10^{-6}$ for the energy equation. Moreover, to evaluate the possibility of contaminant dispersion under the effect of fan operations, a CO source of 0.05 g/s was included at the working face for CFD analysis.

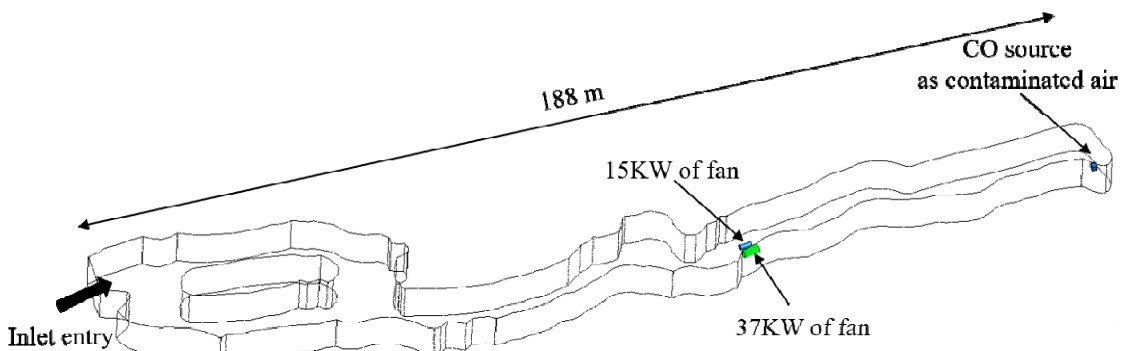

**Figure 14.** 3D geometry layout for CFD analysis.

**Table 4.** Initial conditions of the CFD simulation.

| Parameters | CFD Model |
| --- | --- |
| Inlet boundary | Pressure inlet |
| Wall boundary | Friction wall |
| Ventilation resistance ($k$) | 0.014 kg/m$^3$ |
| Wall temperature | 20 °C |
| Mesh type | Tetrahedron elements |
| Solution model | Turbulence model ($k - \varepsilon$) |
| Mesh size function | Proximity and curvature |
| Number of mesh elements | 500,000 |
| Simulation condition | Transient-state conditions |

For the CFD simulation, a 15 kW low-pressure fan and a 37 kW high-pressure fan were installed at the same location as in the site tests. All input information for CFD simulation was collected based on the mine site study. The fan specifications summarized in Table 1 were applied in CFD analysis and Table 5 shows the fan input data. For the purpose of comparison, both Scenario I and II in the site study were simulated in the CFD analysis. In Scenario I, 15 kW and 37 kW fans were operated in blowing mode, while in Scenario II the 15 kW low-pressure fan was operated in exhaust mode. The pressure profiles and differential pressures were compared.

**Table 5.** Fan characteristics in the CFD analysis.

| | Fan Type | High-Pressure Fan | Low-Pressure Fan |
|---|---|---|---|
| Fan dimension | Discharge diameter (m) | 1.4 | 0.95 |
| | Length (m) | 3.0 | 2.2 |
| Power (kW) | | 37 | 15 |
| Fan efficiency (h) | | 0.7 | |
| Fan Pressure (Pa) | | 551 | 235 |
| Outlet velocity (m/s) | | 30.6 | 23.5 |

*4.2. Discussion and Comparison of the Site Study with the CFD Analysis*

Figure 15 shows the pressure profiles of the CFD analysis. It can be clearly observed that the high-pressure zone was distributed between the fan location and the working face in Scenario I and II. In Scenario I, due to the effects of two blowing fans, the pressure differentials were relatively higher than in Scenario II. The differential pressure in Scenario I estimated by CFD analysis was found to be approximately 30 Pa as shown in Figure 15a. A similar result can be obtained in Scenario II, as plotted in Figure 15b, which shows a pressure differential of approximately 15 Pa. Since the operation of the 15 kW fan was reversed, the differential pressure between the fan front and rear was decreased.

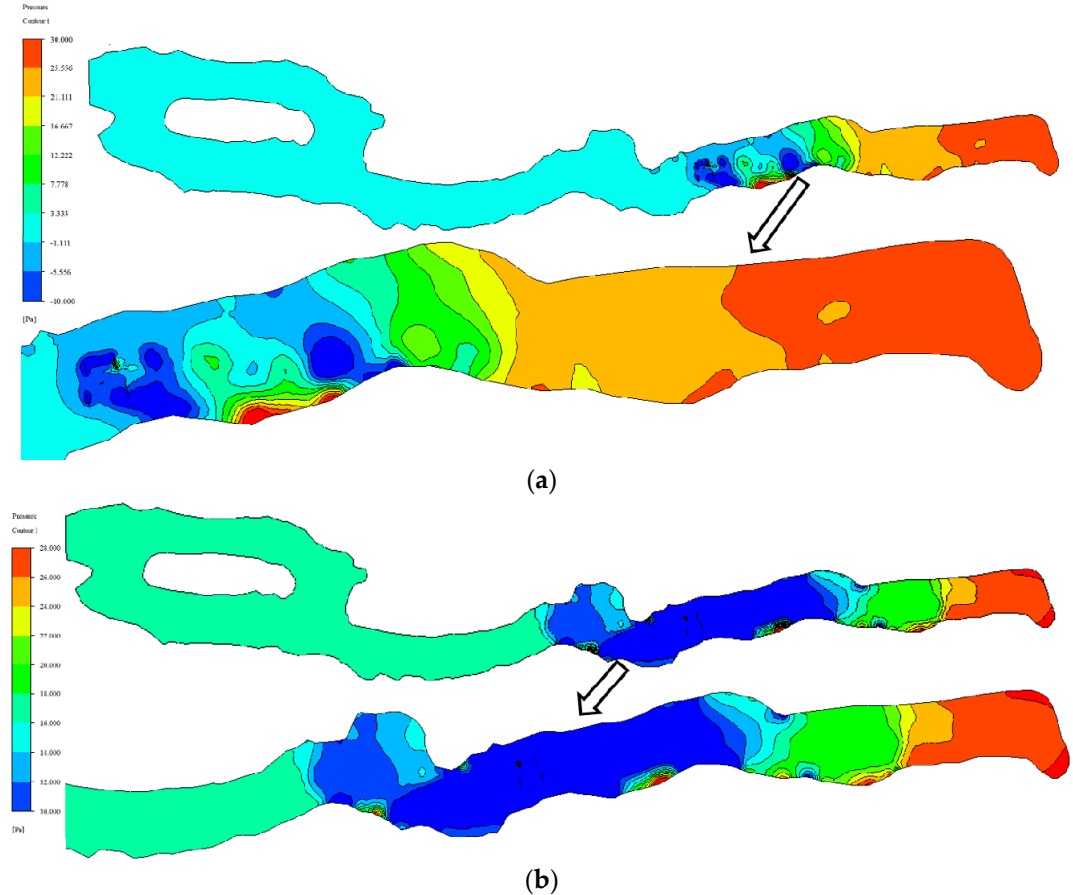

**Figure 15.** Pressure profiles by CFD analysis. (**a**) Scenario I; (**b**) Scenario II.

Figure 15 shows that the whole working space located downstream of the fan is pressurized. Additionally, the site test results indicated the airflow within this pressurized zone was vigorously recirculated. To prove this, the air velocity contours demonstrated in the CFD analysis are plotted in Figure 16. It is clearly seen in Figure 16a that the jet streams discharged from the two blowing fans

lose most of the momentum when they collide with the entry wall. Especially, the high discharged air velocity of the 37 kW fan of high-pressure has collided directly with the curved wall, 30 m from the fan location. In Figure 16b of Scenario II, the same jet stream collision occurs even for the 15 kW fan in exhausting mode. In brief, after the collision, due to this collision, the air was moving at extremely low velocity and eventually sucked into the fan again. This led to the recirculation illustrated clearly in Figure 16. This recirculation is the main reason for the observation made in the site study that most of the airflow remains trapped within the pressurized zone during fan operation.

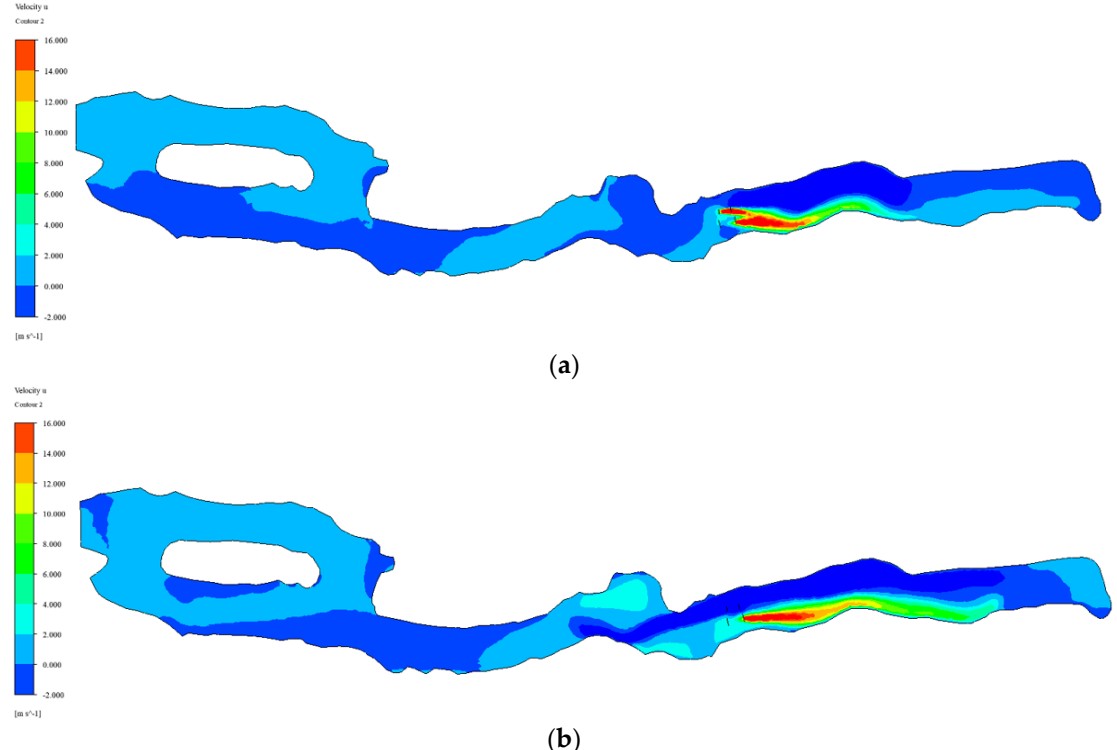

**Figure 16.** Air velocity contours by CFD analysis. (**a**) Scenario I; (**b**) Scenario II.

Figure 17 illustrates the differential pressure between high- and low-pressurization zones shown in the CFD results after 60 min of simulation time. After the fans were turned on, a high-pressurization zone was created in both scenarios. The average differential pressure was 29.23 and 16.15 Pa for Scenario I and II, respectively. These results are very similar to those obtained in the site experiments as shown in Figure 7b,d,f of Scenario I and Figure 11b,d of Scenario II. Consequently, the site experiments show, and the CFD analysis confirms, that by the operation of two fans, a positively pressurized zone can be created continuously near the face to prevent gas leakage from the backfilled site.

To compare the experimental and numerical results in more detail, the average pressures by distance from the face were plotted in Figure 18. The two sets of data from the site experiments and the CFD studies are extremely similar; in Scenario I, a highly pressurized zone can be observed as far as 40 m from the working face as in Figure 18a, while a low-pressure zone is located from 60 to 140 m away from the working face. A similar pattern can be found in the results of Scenario II as shown in Figure 18b. The comparisons are summarized in Table 6.

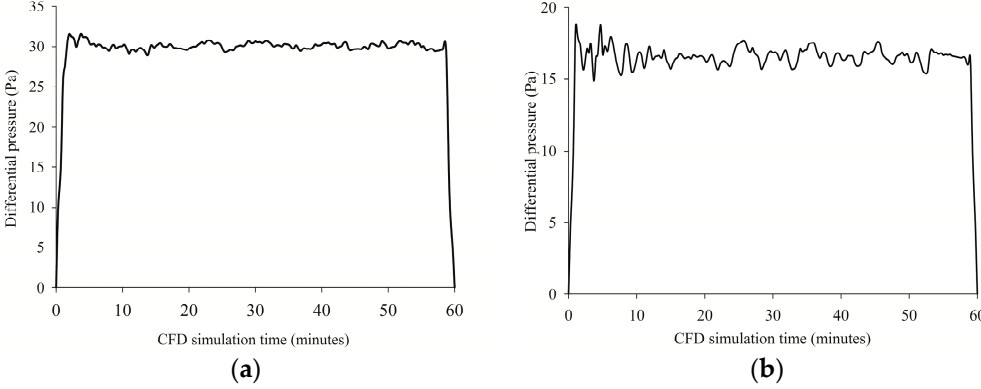

**Figure 17.** Differential pressures by CFD results. (**a**) Scenario I; (**b**) Scenario II.

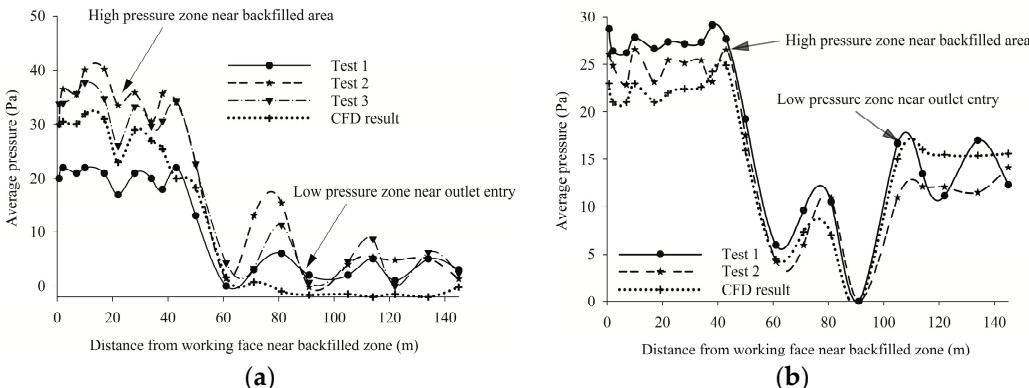

**Figure 18.** Average pressures by distance between CFD and experimental results. (**a**) Scenario I; (**b**) Scenario II.

**Table 6.** Comparisons between the site experiments and the CFD analysis.

| Scenarios of Experiment | | Differential Pressure by Experimental Result (Pa) | Differential Pressure by CFD Results (Pa) |
|---|---|---|---|
| Scenario I (15 kW and 37 kW of blowing fans operation) | Test 1 | 22.3 | 29.23 |
| | Test 2 | 32.78 | |
| | Test 3 | 30.5 | |
| Scenario II (15 kW of exhausting and 37 kW of blowing fan) | Test 1 | 15.38 | 16.15 |
| | Test 2 | 17.56 | |

### 4.3. Possibility of Contaminant Dispersion

The pressurizing ventilation can be designed to have twofold effects. Once a strong airstream discharged from the fans collides with the working face, then the positively pressurized boundary created on the face will deter the leakage of gases entrapped and adsorbed in the backfilled zone pore space. In addition, even though some gas is leaked, it can be confined within the pressurized zone between the face and fan installation. This will minimize workers' exposure. The first effect can hardly be evaluated since the pore pressure is not known. However, in this paper, the second effect was studied by assuming a gas source at the face having an emission rate of 0.05 g/s. This emission rate is similar to that of the methane emission rate at an abandoned coalbed methane wells reported by Johnson and Heltzel [19]. A series of CFD analyses were carried out to account for the gas dispersion behavior during the fan operation.

Figure 19 shows the CO gas concentration profiles by CFD analysis after 20, 40, and 60 min of simulation time. It can be observed that the leaked-out CO was dispersed mainly between the fan location and the working face in Scenario I and II. Especially in Scenario I, due to the effects of two blowing fans, the pressure differentials between the pressurized zone and the outside are relatively higher than in Scenario II. As a result, less CO was dissipated outside the pressurized zone, and this resulted in higher CO concentrations at the space within 30 m from the face after 20 min of fan operation as shown in Figure 19a. In Scenario I, it is well demonstrated that CO leaked out of the backfilled zone can be efficiently confined to the small space within approximately 30 m of the face. Even when the smaller 15 kW fan was reversed in Scenario II, a single 37 kW fan seems to be effective to pressurize the working space and control the dispersion of CO outside the working area as shown in Figure 19b.

Figure 20 illustrates the CO gas concentration variations at three monitoring stations, Point 1, 2, and 3, during the 60-min simulation. The three stations are 2, 40, and 150 m away from the face, respectively. Due to the two blowing fans in Scenario I, the concentrations at Point 1 fluctuate more significantly than in Scenario II. This fluctuation close to the face does not influence the workers' exposure. Even though the CO level at Point 2 continued to increase linearly during the CFD analysis, it was 8.3 and 7.3 ppm after 60 min in Scenario I and II, respectively. If the linear trend was assumed to continue even after 60 min, the CO levels at Point 2 would have been 16.6 and 14.6 ppm after 2 h, still far less than 30 ppm which is the 8 h TWA permissible level of CO in Korea. These results imply that due to the pressurized zone created by two blowing fans installed as in Scenario I and II, the gases leaked from the backfilled zone can be well-confined near the working face and thus minimize the workers' exposure.

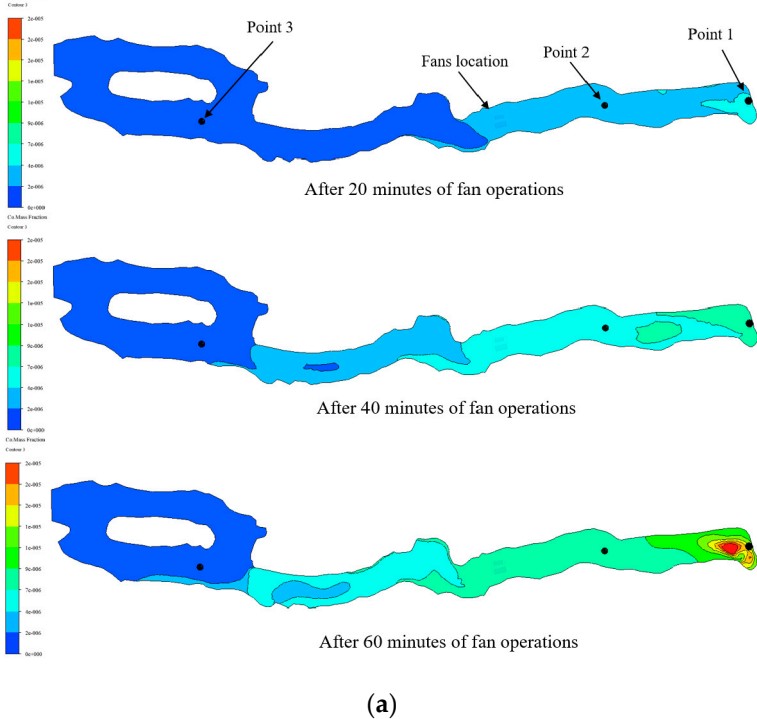

After 20 minutes of fan operations

After 40 minutes of fan operations

After 60 minutes of fan operations

(**a**)

**Figure 19.** *Cont.*

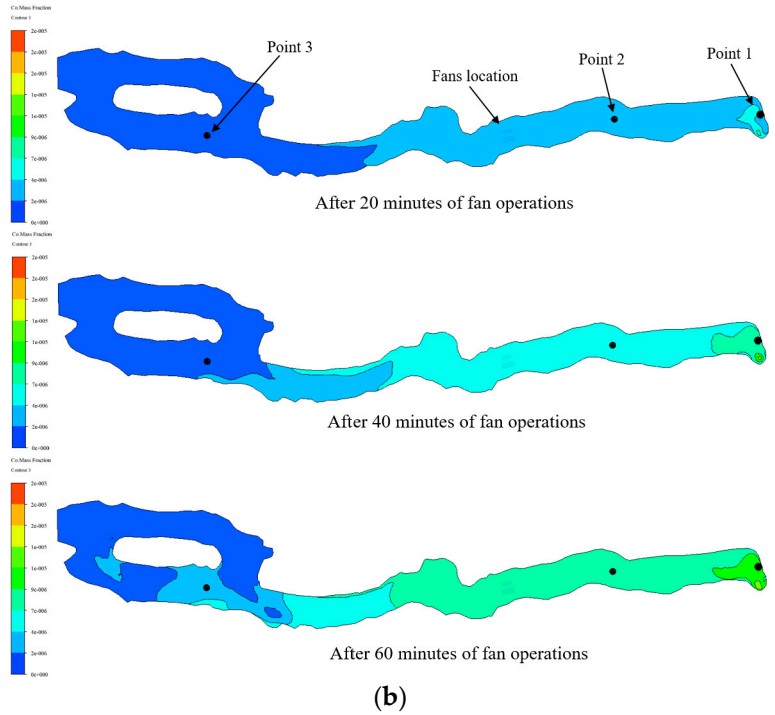

**Figure 19.** CO gas concentration profile by CFD analysis. (**a**) Scenario I; (**b**) Scenario II.

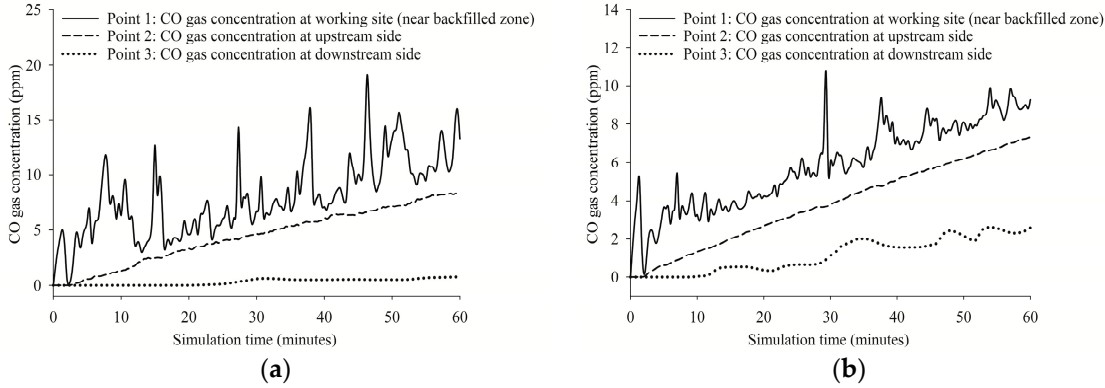

**Figure 20.** CO gas concentration at working face, upstream side, and downstream side of airway. (**a**) Scenario I; (**b**) Scenario II.

## 5. Conclusions

The so-called pressurization ventilation technique was evaluated as a ventilation scheme to control the leakage and dissipation of hazardous gases from the working site backfilled by fly-ash-based materials. The method originally developed for securing a safe escape route in a building fire was designed to create a certain level of pressure differentials between the fire zone and the escape paths. The test site for this site was a limestone mine development heading with a large cross-sectional area (8 m (W) × 7 m (H)). To pressurize the area within the vicinity of the working face and thus confine the contaminated zone to minimize the workers' exposure, two different types of fans—15 kW and 37 kW jet-fan-type ventilation fans—were developed and several ventilation scenarios were tested at the test site and also by CFD analysis. Several important results are summarized as follows:

1.  With two fans in blowing mode, a positively pressurized zone can be generated continuously near the face, and the pressure differentials between the downstream and upstream measured

by 40 pressure sensors communicated by CAN ranged between 22.3 and 32.78 Pa. The pressure differential simulated by CFD analysis was 29.23 Pa.

2. With the 37 kW fan in blowing mode and the 15 kW fan in exhausting mode, relatively smaller pressure differentials of 15.38–17.56 Pa were observed and are comparable to the CFD analysis result of 16.15 Pa.

3. Since the differential pressure of 50 Pa specified in British standards to control building fires which have a relatively larger dissipation force than the gas leakage in the mining face, the pressure difference of approximately 30 Pa created by two blowing fans and 16 Pa by one-blowing and one-exhausting fan in this study seems to be sufficient to control the gas leakage and dispersion within the working space.

4. The above conclusion was supported by the velocity distributions measured at both within and outside of the pressurized zones. The air in the pressurized zone was vigorously circulated, while the outside airflow was almost stagnant. This implies that contaminated air can be well confined within the pressurized zone near the face.

5. Since most of the limestone mines in Korea are developed within the steeply dipping veins, the developed entries are not straight but curved irregularly. This makes the jet stream discharged from the fan collide with the nearby sidewalls and reduce the fan efficiency considerably. Therefore, to install the fans for this ventilation scheme, the fan location must guarantee the minimum loss of jet stream momentum in the downstream side.

6. Even though the time required for curing depends on the type of backfilling materials, this ventilation system can be turned on only during the curing period to have a high possibility of containing gas leakage. In addition, it was shown that the pressurization system with low-pressure fans can be operated at low cost. These are the economic advantages of the system discussed in this study.

7. With this pressurization system, the positively pressurized zone can be generated continuously near the face to restrain the gas leakage and also confine the leaked-out gases near the backfilling face. However, the efficiency of preventing gas leakage from the backfilled zone was not analyzed in this study, due to the lack of basic knowledge about the permeability characteristics of backfilled zones. The total efficiency of the system is a topic requiring further study in the future.

**Author Contributions:** Data collection and experimental works: V.-D.N., W.-H.H., R.K.; Writing, discussion, analysis: V.-D.N., C.-W.L.

**Funding:** This research was funded by the Ministry of Science and ICT (MSIT), the Ministry of Environment (ME) and the Ministry of Trade, Industry and Energy (MOTIE), grant number 2017M3D8A2090024 under the National Strategic Project-Carbon Upcycling Program of the National Research Foundation of Korea (NRF).

**Acknowledgments:** The authors would like to thank to Daesung Mining Development INC, S. Korea; the engineers and leaders of DFC large-opening underground limestone mine, Chungbuk province, S. Korea for their help and cooperation.

**Conflicts of Interest:** The authors declare no conflict of interest.

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
