# Peer review of "Pressurization Ventilation Technique for Controlling Gas Leakage and Dispersion at Backfilled Working Faces in Large-Opening Underground Mines: CFD Analysis and Experimental Tests"

_sustainability, doi:10.3390/su11123313_

Round 1

Reviewer 1 Report

The presented idea of using high pressure of air in mine tunnels for protection of backfilled spaces against leakage seems to be good, but proposed methods doesn’t look useful.

The reason is that pressurisation systems used in the stairs cases are used only occasionally, but in the described problem, the system should be working continuously, what seems to be very expensive. More over, the general assumption of pressurised systems is to supply the air into the space very hermetic, when the mine tunnel is always open on one side. The jest fun systems are rather longitudinal ventilation systems and even the effect of higher pressure is achieved, also the continues air flow is in the proposed method crated, which would increase dispersion of the contamination.

In my opinion the paper should be corrected by adding simulation results of contamination concentration in the tunnel and vector velocities in a few cross sections of the tunnel, what probably will show that the pushed air returns to the beginning of the tunnel.

Also economical aspect of the proposed method should be presented.

Author Response

Response 1: We appreciate all your comments on our work very much. Point 1 in your review: High cost of fan operation. However, as mentioned in the abstract and introduction part, backfilling materials are developed as slurry compounds of fly ash and other carbonates minerals which behave just like cement mortar. Therefore, the pressurizing fans system will be turned on only during the curing period of the backfilling materials which is expected to be 1-2 days at most.  This will be explained in the revised paper.

Response 2: Thank you very much for your comment. The CFD analysis results of contaminant dispersion are added in Section 4.3 of the revised paper. The workers will be at the backfilling site during the backfilling operation. Thus, the purpose of this ventilation method is to confine the contaminants (mostly gases leaked out) near the working face to minimize the workers’ exposure. The working site is a blind section. CFD analysis of the gas dispersion by FLUENT had been carried out and the CO concentration profiles are added to the revised paper. The results imply that due to high pressurization zone created by two blowing fans in Scenario I, CO leaked from the face can be well confined within the pressurized zone near the working face. In Section 4.3, discussion about the possibility of contaminant dispersion is included from line 304 to 339 in the revised paper. Thank you very much.

Response 3: We appreciate very much your comment. However, there are several reasons the economical aspect was not included in this study. Firstly, as mentioned in the abstract of our manuscript, the pressurization ventilation techniques originally designed to control building fire had never been applied to the mines. Therefore, the fundamental aim of this study is to evaluate the applicability of pressurization ventilation at the backfilled working face in large-opening underground limestone mines. The results of experimental and numerical showed the efficiency of the ventilation method; the positively pressurized zone can be generated continuously near the face to restrain the gas leakage and confine the leaked-out gas near the backfilling face. In addition, a low-pressure low-power 15KW fan was developed for this study. Lower power requirement leads to less operation cost. Since this is the first study to control the toxic gas leakage from the backfilled zone by pressurization ventilation system in mine, so there was no previous method that has been used to compare economical efficiency. Thank you very much.   

Reviewer 2 Report

The manuscript "Pressurization Ventilation Techniques for Controlling Gas Leakage and Dispersion at Backfilled Working Faces in Large-Opening Underground Mines: CFD Analysis and Experimental Tests" reports an interesting analysis on the application of the pressurization ventilation technique in underground mines. Both experimental tests and CFD simulations were used to support discussion and results. The manuscript is interesting and I recomment its publication.

In order to even improve it I suggest to consider the following issues:

- please consider to change the "-" symbol in table 2 for list of elements with another symbol or to align left column 2.

-  Since do you have a validated CFD model, why do not use it to check the gas leakage and dispersion control? nevertheless, if this is out of your aims, please consider to include a larger discussion on this point in section 4, being directly connected with the objective of the adoption of pressurization ventilation techniques in mines. By this way you may better support conclusion point 4 .

Author Response

Response 1: Thank you very much for your excellent comment. I already changed the Table 2 as your comment. Thank you.

Response 2: Thank you very much for your comment. The CFD analysis results of contaminant dispersion are added in Section 4.3 of the revised paper. The workers will be at the backfilling site during the backfilling operation. Thus, the purpose of this ventilation method is to confine the contaminants (mostly gases leaked out) near the working face to minimize the workers’ exposure. The working site is a blind section. CFD analysis of the gas dispersion by FLUENT had been carried out and the CO concentration profiles are added to the revised paper. The results imply that due to high pressurization zone created by two blowing fans in Scenario I, CO leaked from the face can be well confined within the pressurized zone near the working face. In Section 4.3, discussion about the possibility of contaminant dispersion is included from line 304 to 339 in the revised paper. Thank you very much.

Round 2

Reviewer 1 Report

I confirm that most of my remarks are implemented, however, the economic aspect of the proposed method, which was asked to be shown is finally not presented at all. I suggest to mention this element in conclusions because it could be useful for them, who could take into account using such a method as described in their real mines. Also, the limitations of the proposed system should be very clearly described.

Author Response

Point 1: I confirm that most of my remarks are implemented, however, the economic aspect of the proposed method, which was asked to be shown is finally not presented at all. I suggest mentioning this element in conclusion because it could be useful for them, who could take into account using such as a method as described in their real mines. Also, the limitation of the proposed system should be very clearly described.

 Response 1: We appreciate all your comments on our work very much. Two more conclusion point was added in conclusion part as you mentioned to describe clearly potential limitation and economical aspect of our study from line 274 to 283 of the manuscript. Thank you very much.

        6. Even though the time required for curing depends on the type of backfilling materials, this ventilation system can be turned on only during the curing period having the high possibility of gas leakage. In addition, it was shown the pressurization system with low-pressure fans can be operated at low cost. These are the economic advantages of the system discussed in this study.

        7. With the pressurization system, the positively pressurized zone can be generated continuously near the face to restrain the gas leakage and also confine the leaked-out gases near the backfilling face. However, the efficiency of preventing gas leakage from the backfilled zone was not analyzed in this study, due to the lack of basic knowledge about the permeability characteristics of the backfilled zone. The total efficiency of the system is the topic requiring further study in the coming time.  
